# PCBs in Chinstrap Penguins from Deception Island (South Shetland Islands, Antarctica)

**DOI:** 10.3390/toxics13060430

**Published:** 2025-05-24

**Authors:** Miguel Motas, Silvia Jerez-Rodríguez, José Manuel Veiga-del-Baño, Juan José Ramos, José Oliva, Miguel Ángel Cámara, Pedro Andreo-Martínez, Simonetta Corsolini

**Affiliations:** 1Department of Toxicology, Faculty of Veterinary, Regional Campus of International Excellence “Campus Mare Nostrum”, University of Murcia, Campus of Espinardo, 30100 Murcia, Spain; motas@um.es (M.M.); silviajerez@um.es (S.J.-R.); 2Department of Agricultural Chemistry, Faculty of Chemistry, Regional Campus of International Excellence “Campus Mare Nostrum”, University of Murcia, Campus of Espinardo, 30100 Murcia, Spain; chemavb@um.es (J.M.V.-d.-B.); josoliva@um.es (J.O.); mcamara@um.es (M.Á.C.); 3National Centre for Environmental Health (CNSA), Instituto de Salud Carlos III (ISCIII), 28220 Madrid, Spain; jjramos@isciii.es; 4Department of Physical, Earth and Environmental Sciences, University of Siena, Via Mattioli, 4, 53100 Siena, Italy; simonetta.corsolini@unisi.it

**Keywords:** Antarctica, krill, PCBs, *Pygoscelis antarticus*, TEQs

## Abstract

The aim of this study was to evaluate the concentration of polychlorinated biphenyls (PCBs) in chinstrap penguins (*Pygoscelis antarctica*) and krill (*Euphausia superba*) from Deception Island (South Shetland Islands, Antarctica) to provide additional data of the PCB presence in Antarctica. To this end, 34 samples of different tissues corresponding to four adult specimens and six chicks, and krill from the area were studied. The selected samples were analyzed for the determination of 27 congeners of PCBs by gas chromatography. Adult specimens accumulated PCBs mainly in the liver (33%, 1330.82 ± 733.69 pg·g^−1^ wet weight, w.w.) and muscle (25%, 1029.73 ± 823.4 pg·g^−1^ w.w.), whereas the brain showed the highest levels in chicks (36%, 1215.83 ± 955.19 pg·g^−1^ w.w.). Regarding krill, our results were five to eight times lower than the levels found in krill from King George Island and from the Ross Sea. Further, a distribution analysis of PCBs in penguins according to Regulation 2013/39/UE and Commission Regulation (EU) No 277/2012 was also performed, and PCBs were categorized into three groups (dioxin-like-mono-*ortho*, non-dioxin-like-indicators, and others-non-dioxin-like). The data indicate that the content of the other group was generally higher than that of the other two PCB groups for both adults and chicks. Notably, the liver consistently exhibited the highest proportion of the other group.

## 1. Introduction

Antarctica has long been regarded as one of the few pristine regions on the planet and serves as a global symbol of conservation. Its isolation, created by natural barriers such as water masses and atmospheric currents, has contributed to this perception. However, the idea that this area is completely unpolluted began to be challenged in the 1960s. Sladen et al. [1] were among the first to identify the pesticide dichlorodiphenyltrichloroethane and some of its related compounds in tissue samples of Adelie penguins and a Crabeater seal collected in Antarctica in 1964. This was notable because pesticides had never been used on the continent. Since then, many other persistent organic pollutants (POPs) have been identified in Antarctic samples, indicating that even the most isolated part of the planet is not immune to contamination and its global effects [2,3,4]. In addition, POPs may also be transported to these regions via long-range atmospheric transport and oceanic transport, although the amount of transported POPs seems much lower, to a significantly lesser extent, via pelagic organisms and migratory birds [5].

The phenomenon involving the accumulation of POPs and other contaminants in polar regions and remote areas is referred to as global distillation. This process occurs due to the mechanisms of cold condensation and global fractionation [6]. As a result, research into the abundance and behavior of environmental pollutants in Antarctic ecosystems has gained significant interest within the international scientific community in recent years [7].

POPs present a significant global pollution issue because they can be carried over long distances, primarily through atmospheric currents. These pollutants can reach the polar regions, where low temperatures and prolonged periods of darkness allow them to degrade very slowly, leading to their accumulation in ice [8]. When the ice thaws, these pollutants are released back into the environment, where they can enter food webs, accumulate in the tissues of organisms, and biomagnify [9]. Furthermore, climate change and rising temperatures in certain areas, such as the Antarctic Peninsula, may intensify the transport and deposition processes of these pollutants. Among these substances, polychlorinated biphenyls (PCBs) are particularly noteworthy for their persistence and ability to accumulate. It is well-established that most PCB congeners, which have been used for decades, have been transported on a large scale and have reached Antarctica [2].

PCBs are chlorinated aromatic hydrocarbons that are synthesized from biphenyl. There are 209 identified congeners, which share the general formula C_12_H_10_−nCl_n_ (where 1 ≤ n ≤ 10). As a class of POPs, also referred to as semi-volatile organic compounds, PCBs exhibit heat resistance and excellent electrical insulation properties. However, these pollutants can negatively impact human health even at low exposure levels [10]. Despite a ban on their production, their chemical stability allows PCBs to persist in the environment and its degradation takes more than 40 years [11,12].

Exposure to PCBs is linked to neurotoxicity, endocrine dysfunction, and reproductive disorders. Specifically, PCBs can be chemically and enzymatically converted into toxic quinone-type derivatives. These quinones act as Michael acceptors, meaning they can react with glutathione, proteins, and DNA, leading to cellular damage. Additionally, quinones possess redox properties that can increase oxidative stress, among others [13].

In general, sea birds play a significant role in environmental biomonitoring due to their widespread distribution, relative ease of identification, and crucial position within aquatic ecosystems. They are particularly sensitive to changes caused by human activity in their habitats. By studying their diets, researchers can monitor shifts in the abundance of their prey. These shifts provide insights into a range of issues, including changes in climatic conditions, the impact of human exploitation on certain species, and variations in nutrient intake, which can be a primary pathway for contaminants to enter their bodies [14,15].

Antarctic penguins have been suggested as potential sentinels for monitoring global environmental pollution. They possess several characteristics that make them effective sentinels in other regions. As top predators and long-lived species, they are subject to biomagnification and bioaccumulation. Additionally, they have wide distribution ranges with abundant populations and a large body size that facilitates sampling, and they can integrate contamination over both time and space [16]. Furthermore, the various penguin species typically feed on krill (*Euphausia superba*), followed by fish and squid [17], and monitoring their diet can also provide insights into the bioaccumulation of contaminants.

On the other hand, a search for publications on PCBs in penguins in the WoS scientific database using the keywords PCBs (Topic) and penguin * (Topic) yielded 97 publications as of May 2025 (Figure 1).

As can be observed in Figure 1, studies on PCBs in penguins began in 1975, peaking at seven publications in 2017, and then declining to two publications and one publication in 2024 and 2025, respectively. Table 1 shows some of the studies that have been carried out on the possible presence of PCBs in different tissues of various species of penguins in different years and locations.

From the authors’ perspective, there are currently few publications on the subject to evaluate temporal trends, and additional data on these pollutants in the region are essential in order to further investigate the temporal, spatial, or interspecific variability in their presence. Additionally, monitoring and managing the presence and abundance of PCBs in ecosystems enables us to assess the extent and progression of contamination over time. This allows us to anticipate potential harmful effects, make informed decisions regarding management, regulation, and protection, and set priorities for future research [54].

Therefore, the primary objective of this study was to evaluate the concentration of PCBs in chinstrap penguins (*Pygoscelis antarctica*) and krill (*Euphausia superba*) from Deception Island (South Shetland Islands, Antarctica) to provide additional data of the PCBs presence in Antarctica. Moreover, we described the PCBs tissue distribution, and the distribution according to the Directive 2013/39/UE and Commission Regulation (EU) No 277/2012 both in adults and chicks. The relative toxic potential of toxic mono-*ortho* PCBs congeners in penguin tissues was assessed using the Toxic Equivalency Factor (TEF) approach [57], and the Toxic Equivalent (TEQ) concentrations were calculated in adult liver and chick brain samples.

## 2. Materials and Methods

The presence and tissue distribution of PCBs were determined in chinstrap penguins (*Pygoscelis antarctica*) and krill collected during the austral summer season of Antarctic campaigns from 2007 to 2010 from the population located on Deception Island (South Shetland Islands, Antarctica, 63°00′ S 60°40′ O) (Figure 2). On Deception Island, there are currently two summer scientific stations, including the Spanish Gabriel de Castilla and the Argentinian Deception scientific stations, as well as ruins of other stations that were destroyed by the last volcanic eruption in 1967. This island is the crater of a horseshoe-shaped volcano, whose flooded caldera forms a natural harbor unique to the area [58]. This natural refuge has hosted human activities since the beginning of the last century, mainly whale and seal hunters and whaling industries until 1967 [59], and currently supports heavy traffic of ships and cruise ships. The penguin colony where sampling was carried out is Vapour Colony (63°00′ S 60°40′ W), which has an estimated population of 20,000 pairs [60]. To our knowledge, only one article was published on contamination in eggs from this colony.

The carcasses of adults and chicks were collected during the Antarctic campaigns from 2007 to 2010, and preserved frozen at −20 °C in individual polyethylene bags until their analysis the same year. A total of 34 penguin samples were analyzed: liver, kidney, muscle, heart, and brain from 4 adult and 6 chick specimens were studied. Samples of krill, main prey of penguins, were analyzed and obtained from the stomach contents of the penguin specimens, using the stomach lavage method [61,62] (Table 1). Due to the low amount of chick samples, some tissues were homogenized to obtain 5 g pools for the analyses (Table 2).

The selected samples were extracted and then analyzed for the determination of 27 polychlorinated biphenyl congeners (PCBs). The PCB extraction method is based on a previously described method elsewhere [23,63]. Briefly, 4–5 g of each sample was homogenized with sodium sulphate and spiked with 50 ng of PCBs 30 and 209 (Supelco Inc., St. Louis, MO, USA) as internal standards. They were Soxhlet-extracted with methylene chloride and hexane (3:1, 400 mL; 12 h) and the extract was evaporated in a rotary evaporator. An aliquot of each extract was then taken to determine the fat content of each tissue by gravimetry. Possible interferences were eliminated by purification with a Power-Prep system. The samples were analyzed by gas chromatography (Perkin Elmer mod. Autosystem) equipped with a 63Ni electron capture detector (GC-ECD) and a DB-5 capillary column (Supelco Inc., 30 m × 0.25 mm i.d × 0.25 μm). The chromatographic conditions can be found elsewhere [64].

The injection volume was 2 mL. Recovery rates for PCB congeners were evaluated by adding known amounts (25, 50, 100, and 250 ppb; internal standard volume ¼ 200 mL) of PCB congeners (CBs 153, 138, 170, 194, 101, 118, and 156) to a set of samples (n ¼ 6) prior to the analyses. Recovery rates were PCB138 ¼ 97 ± 12%; PCB153 ¼ 93 ± 18%; PCB170 ¼ 86 ± 19%; PCB194 ¼ 92 ± 14%; PCB101 ¼ 97V15%; PCB118 ¼ 87 ± 19%; and PCB156 ¼ 93 ± 12%. The standard solutions used for identification and quantification of single chemicals and for the recovery rate experiments were obtained by Supelco, Inc. (Sigma-Aldrich, St. Louis, MO, USA).

The detection limits (LODs) were calculated as the mean value of each compound in the blanks + 3SD and were, in pg·µL^−1^, as follows: 1.907 (PCB 28), 0.199 (PCB 52), 1.842 (PCB 37), 0.012 (PCB 95), 1.426 (PCB 101), 0.002 (PCB 99), 0.012 (PCB 110), 0.520 (PCB 151), 0.011 (PCB 149), 1.066 (PCB 123), 0.427 (PCB 118), 0.003 (PCB 114), 0.011 (PCB 146), 0.462 (PCB 153), 2.541 (PCB 105), 0.010 (PCB 138), 0.430 (PCB 187), 0.003 (PCB 183), 1.093 (PCB 128), 0.010 (PCB 167), 0.003 (PCB 177), 1.845 (PCB 156), 0.010 (PCB 157), 0.003 (PCB 180), 0.003 (PCB 170), 0.007 (PCB 189), and 0.007 (PCB 209). Blanks were run with each set of 5 samples.(1)$$BMF=\frac{PCBconc\_pred}{PCBconc\_prey}$$

The biomagnification factor (BMF) was assessed using Equation (1):

Where PCBconc is PCB concentration, pred is predator. The BMF was assessed using the concentrations detected in the penguin liver and whole krill specimens. The brain/liver/muscle was used because it is the organ of accumulation; penguins feed on whole krill, so it is appropriate to use the concentration in whole samples.

The relative toxic potential of most toxic PCBs, namely, eight mono-*ortho* substituted congeners mono-*ortho* (PCB 105, PCB 114, PCB 118, PCB 123, PCB 156, PCB 157, PCB 167, and PCB 189), was assessed using the Toxic Equivalency Factor (TEF) approach [65]. TEF values [57] were used to calculate the Toxic Equivalents (TEQs). TEQs were calculated on using mono-*ortho* congener concentrations expressed as pg·g^−1^ lipid wt.

For data analysis, half the value of the detection limit (DL/2) was assigned to samples that did not show detectable levels of contaminants. The results are shown as mean ± standard deviation (SD) and minimum maximum on a wet weight (w.w.). After testing the normality and homogeneity of variances, a non-parametric analysis was realized. To evaluate significant differences between concentration of PCBs, in a general way, a Youn’s t-test was applied, which were considered statistically significant at *p*-value ≤ 0.05. To verify the existence of significant differences between PCBs for each of the tissues, a Mann–Whitney test was performed where a U-value < Mann–Whitney U was tabulated, and a *p*-value ≤ 0.05 value is considered significant. Further information about statistical analysis can be found elsewhere [66,67].

Data analysis and graph generation were performed using the R studio IDE Desktop 2024.09.01 tool and R 4.4.2 [68]. Different R libraries were used depending on the analysis performed: for example, the readxl library was used to import data into R, the ggplot2 was used for graph analysis, and Paired Data was used for robust statistical. The code for the R script is available at the request of the authors.

## 3. Results and Discussion

### 3.1. PCB Levels and Temporal Trend in Pygoscelis Antarctica

The analyses revealed the presence of PCBs in chinstrap penguins and krill from Deception Island (Table 3). These results are consistent with those expected; several studies have reported that, despite its prohibition and due to its persistence, high concentrations of PCBs continue to be detected in all environmental compartments in remote regions such as Antarctica [69], and the existence of long-range atmospheric transport of PCBs from South America to the Antarctic Peninsula area has been reported [70]. These compounds were already detected at the end of the 1960s in egg samples of Antarctic penguins from the Ross Sea region [8,40]. More than forty articles have studied the abundance of POPs in *Pygoscelis* penguins (mainly in blood, eggs, feces, and internal tissues) [16,71] and data are still scarce and heterogeneous in some Antarctic sectors and species; for instance, chinstrap penguin is the species less studied [72].

Figure 3 shows the distribution of all PCBs between adult penguins and chicks.

Since the groups shown in Figure 3 do not have the same distribution shape and do not meet the assumptions of normality and equal variances, Youn’s test [73] was used. This test does not require any specific assumptions and can accommodate outliers. The *p*-value for the groups was 0.0042, which is lower than the 0.05 threshold, indicating significant differences in the PCB concentration between adults and chicks.

Due to the small number of samples from each tissue analyzed for adult and chick penguins, it was not possible to ensure the normality and homogeneity of variances of the results obtained for each tissue. Comparisons among the concentrations of ∑PCBs in different tissue analyzed both in adult and chick penguins together with PCB krill levels can be observed in Figure 4. Non-significant differences were found in these comparisons (U-value > U Mann–Whitney and *p* > 0.05) due to the high dispersion of values between the same type of tissue, as shown by the standard deviation in Table 3; therefore, the results in Figure 4 can only be interpreted at a descriptive level. However, this figure shows similar patterns. ∑PCBs in the liver, muscle, and heart were 25–52% lower in chicks than in adult specimens. On the contrary, in the kidney and brain, the ∑PCBs were 18–47% higher in chicks than in adults. Regarding the ∑PCBs level in krill, as expected, it was 14–83% lower than the levels detected in penguin tissues and these results are in agreement with those previously reported for PCBs in other areas in Antarctica [9,30]. The krill sample analysis allowed estimating the biomagnification factor (BMF = PCB concentration in predator/PCB concentration in prey) which resulted in a maximum level of 5.85 (PCBs in penguins’ liver/PCBs in krill). This BMF was slightly higher than those found by Cipro, Taniguchi, and Montone [30] between *Pygoscelis* penguins and krill from King George Island (2.10–3.02) but much lower than those found in emperor penguin liver and krill samples (61.3–102) [46].

It is difficult to establish temporal trends of PCB concentrations in the study area because there are very few previous data available for the chinstrap penguin, and even fewer if we consider the geographical area studied in the present study, i.e., Deception Island, although an increasing trend of PCB concentrations in the eggs of the chinstrap penguin from the nearby King George Island from 1993 to 2005 (from 830 to 37,300 pg·g^−1^ w.w.) has been described [72]. Despite governments of several countries starting to ban the use and production of PCBs more than two decades ago, this increase could be related to the release from legal or illegal stocks or PCB-containing equipment that are transported to the Antarctic Region through LRAT. Although we analyzed tissues and not eggs, and, thus, this comparison should be cautiously considered, our results on the PCB concentration were one to two orders of magnitude lower than those measured in 2004–2005 in eggs of the chinstrap penguin from King George Island and like those found in 1993 [29,30,55]. These results could reflect a contrary trend of PCB concentration in this Antarctic area. A similar decreasing trend was earlier observed in Arctic seabirds [74]; a delay in the transport of POPs to Antarctica can exist, due to geographical factors and transport pathways, which may also affect the bioaccumulation in organisms [72].

In any case, the PCB level detected in Arctic seabirds are two to three orders of magnitude higher than those found in the present study (e.g., 124,700–448,700 pg·g^−1^ w.w in the blood of great black-backed and glaucous gulls [75]). These relevant differences between the Polar Regions were previously described [75], and are probably a result of the different distances to the pollution sources.

### 3.2. Tissue Distribution

Figure 4 shows the PCB abundance in adult and chick penguin tissues. However, these results showed that adult specimens accumulated PCBs mainly in the liver (33%) and muscle (25%), whereas the brain showed the highest levels in chicks (36%). In general, our results showed lower levels of PCBs than those found in *Pygoscelid* penguins from other Antarctic locations (see Table 1). In the liver, muscle, and brain, the ∑PCBs concentrations were 1130 pg·g^−1^ w.w., 1029 pg·g^−1^ w.w., and 1215 pg·g^−1^ w.w., respectively (Table 3), lower than but in the same order of magnitude as those detected in the blood and eggs of chinstrap penguins from King George Island (mean level: 4500 [23] and 6000 [29] pg·g^−1^ w.w.), and in the eggs of chinstrap penguins from the same localization, i.e., Deception Island (4710 pg·g^−1^ w.w. [16,29]) (see Table 1). The other tissues analyzed in the present study showed levels of PCBs one order of magnitude lower (Table 3). Regarding the levels reported in the eggs of *Pygoscelid* penguins from King George Island (26,000–37,300 pg·g^−1^ w.w.) [30], our results showed PCB levels one or two orders of magnitude lower (see Table 1).

To our knowledge, only one study [16] has assessed PCBs levels in chinstrap penguin tissues in our location. Our samples of the liver, muscle, heart, and brain from Deception Island contained 4–53 times lower levels of PCBs than tissues of chinstrap penguins from King George Island (7150 pg·g^−1^ w.w., 20,498 pg·g^−1^ w.w., 2474 pg·g^−1^ w.w., and 23,074 pg·g^−1^ w.w. in the liver, muscle, heart, and brain, respectively [76]). The different ecological niche of this population could explain these results, there is evidence that even a little spatial variation can lead to a significant variation inter- or intra-specifically [30]. In any case, the comparison of these results should be made with caution due to the limited number of samples included in the different studies (see Table 1).

The liver of chinstrap penguins in our study also showed levels around one order of magnitude lower than those detected in the liver and blood of other Antarctic seabirds such as south polar skuas (11,150 pg·g^−1^ w.w. in liver [25]; and 9000 pg·g^−1^ w.w. in blood by [75]). These differences among seabirds are consistent with those expected since flying seabird species that overwinter north of the Southern Ocean often accumulate higher contaminant levels than penguins that overwinter in the Antarctic seawaters [28,77].

Regarding krill, our results were 5–8 times lower than levels found in krill from King George Island (1129 pg·g^−1^ w.w. [78]; and 12,300 pg·g^−1^ w.w. [30]), from the Ross Sea (1900 pg·g^−1^ w.w. [25]), and from Terra Nova Bay (900 pg·g^−1^ w.w. [40]) (see Table 1).

### 3.3. PCB Fingerprint and Class of Isomer Analyses

Differences among PCB congeners (fingerprints) in adults, chicks, and krill can be observed in Appendix A. The liver was the main organ for PCB accumulation in adult penguins and, in this organ, PCB 37 was the most abundant congener with an average contribution to total PCBs of 24% (Appendix A). In the brains of chicks, where the highest levels of PCBs were found, the congeners PCB 37, PCB 110, and PCB 114 made up 18%, 18%, and 15% of the residue, respectively (Appendix A). Three congeners identified as predominant in our samples also showed high percentages in the total residue of chinstrap penguins from King George Island (PCB 110 and PCB 153 in the liver and PCB 52 in the heart) [76]. PCB 95 has been previously identified [23] as one of the predominant congeners in the blood of chinstrap penguins from King George Island and this congener was also relevant in our results, especially in krill [PCB 95 made up 63% of the residue (Appendix A)]. Regarding the samples of krill coming from King George Island, PCB 99 was identified as a predominant congener [78], in accordance with our results, but PCB 95 was not found, and PCB 187 was less than 2% of the total residue in the mentioned study in contrast to our results.

Unlike the results obtained in the blood of chinstrap penguins from King George Island [23], PCB 138 and PCB 153 were abundant congeners in some of our samples (Appendix A). These congeners are among the most represented in organisms’ tissues because of their persistency: they were also identified as abundant compounds in other studies in seabirds from Antarctica, including the eggs of chinstrap penguins from King George Island [16,29,50], and seabirds from other regions of the world [63,79].

PCB congeners nos. 114, 118, 128, 138, 153, 167, and 180 were the most abundant in our samples of adults and chicks, in agreement with other studies on eggs of chinstrap penguins from Deception Island (more than 50%) and from King George Island (more than 4% of total residue) [16,29]. The finding of similar PCB fingerprints in adults and chicks could confirm the PCB maternal transfer in Antarctic chinstrap penguins, as reported earlier elsewhere [25] for other Antarctic seabirds. PCB congeners with chlorine atoms substituted in the 2,4,5 positions (e.g., penta-CB no. 118, hexa-CB nos. 149, 138, and 153, and hepta-CB no. 180) are the most resistant to the metabolic degradation in fish and invertebrates [80,81], which are preyed on by penguins. Therefore, they are prone to bioaccumulate in fish-eating organisms [39]. Moreover, the detoxifying activity of the P450 cytochrome, and, in particular, CYP3A (specific for PCB detoxification), is significantly lower in Adélie penguins [82]. Similar patterns were already reported in penguins [28].

Tri-CBs, tetra-CBs, and penta-CBs together accounted for more than 60% of the PCB residue in all adult and chick tissues except for the muscle and brain, respectively (Figure 5). These results agreed with those reported in chick chinstrap penguins’ fat: in these samples, low-chlorinated PCB congeners (i.e., tri-tetra- and penta-PCBs) made up a higher contribution (about 80%) [35]. Hexa-CBs were the most abundant congeners in muscle tissue. In chicks, hexa-CBs accounted for more than 60% of the PCB residue in the brain, whereas penta-CBs were predominant in the rest of the tissues (Figure 5B). Krill samples showed a different pattern of abundance and penta-CBs > hepta-CBs were the most abundant class of isomers (80% and 20%, respectively; Figure 5C). Similar results were obtained in the eggs of chinstrap penguin from Deception Island where hexa-PCBs were predominant [16]. On the contrary, it has been reported that tetra-CBs and tri-CBs were predominant in the eggs of chinstrap penguin and krill from King George Island, respectively [30].

It is interesting to note that low-chlorinated PCBs (tri- to penta-PCBs) accounted for >70% of the residue except in adult muscle and chick brain, in agreement with other studies on penguins [83]. This pattern agrees with the preferential long-range atmospheric transport (LRAT) of low-chlorinated congeners to Polar Regions [84].

Finally, as can be observed in Appendix A, not all studies examine the same number or types of PCBs, so it would be interesting for future studies to examine, at least, the indicator PCBs (28, 52, 101, 153, 138, and 180) [85] for comparative purposes.

### 3.4. TEQ Assessment

The relative toxic potential of most toxic mono-*ortho* PCB congeners in penguin tissues was assessed using the TEF approach (Table 4). TEQ concentrations were the highest in the liver of adult penguins, and the brain of chicks. The TEQ values were below the LOD in the liver of adult chinstrap penguins and only PCB 105 was detected (TEQ = 0.004 pg·g^−1^ l.w.). Five mono-*ortho* PCB congeners were detected (PCB 105, PCB 114, PCB 118, PCB 157, and PCB 167) in the brain of chicks and the TEQ value was 0.167 pg·g^−1^. This TEQ value in chick brain samples was one order of magnitude higher than the TEQ value calculated for mono-*ortho* PCB congeners in the eggs and blood of chinstrap penguins from South Shetland Islands [16,23]. The higher TEQ concentrations in the brain could be due to its higher lipid content with respect to blood. High levels in the brain should be monitored because of its importance in all the physiological functions and mainly in chicks that may be more sensitive during the early life stages [86].

### 3.5. Distribution of PCBs in Penguins According to Directive 2013/39/UE and Commission Regulation (EU) No 277/2012

To perform the analysis of the distribution of PCBs analyzed in this study, according to Directive 2013/39/UE [87] and Commission Regulation (EU) No 277/2012 [85], PCBs were categorized into three groups. The first category consists of mono-*ortho*-dioxin-like PCBs, which include PCB-123, PCB-118, PCB-114, PCB-105, PCB-167, PCB-156, PCB-157, and PCB-189. The second category is made up of non-dioxin-like-indicator PCBs, including PCB-28, PCB-52, PCB-101, PCB-153, PCB-138, and PCB-180. The third category encompasses other-non-dioxin-like PCBs, including PCB-37, PCB-95, PCB-99, PCB-110, PCB-151, PCB-149, PCB-146, PCB-187, PCB-183, PCB-128, PCB-177, PCB-170, and PCB-209. The first group is defined according to Directive 2013/39/UE [87], taking into account their TEF [57]. Non-*ortho*-dioxin-like PCBs (PCB-77, PCB-81, PCB-126, and PCB-169), which are also included among the 12 PCBs with TEF along with mono-*ortho*-dioxin-like PCBs, were not analyzed in this study. The second group consist of the six indicator PCBs proposed by Commission Regulation (EU) No 277/2012 [85], while the third group includes non-dioxin-like PCBs analyzed in this study not covered by these regulations.

Figure 6 and Figure 7 show the distribution of the different types of PCBs in the tissues of adult and chick chinstrap penguins in accordance with the above regulations [85,87].

The data show that the content of the other-non-dioxin-like PCB group is generally higher than that of the other two PCB groups for both adults and chicks. Although there are minor variations depending on the specific tissue analyzed, the liver consistently exhibits the highest proportion of other-non-dioxin-like PCBs in both adults and chicks. This observation may be attributed to the liver being one of the most extensively studied organs in scientific literature, particularly regarding its potential for early bioaccumulation [88].

As can also be observed, in the muscle and brain of chickens, the proportion of the mono-*ortho*-dioxin-like group increases in contrast to the decrease in the rest of the PCBs.

The sum of non-dioxin-like-indicator PCBs indicates that, for chicks, the results vary between 41.5 pg ·g^−1^ w.w. and 513 pg g^−1^ w.w. In adults, the range is from 7.1 pg g^−1^ w.w. to 642 pg·g^−1^ w.w.

Likewise, other-non-dioxin-like PCBs not covered by the Directive 2013/39/UE [89,90] and Commission Regulation (EU) No 277/2012 [85] are those that appear mostly in the penguins analyzed. This result is consistent with previously published papers, which state that dioxin-like PCBs are only a minor component of the total PCBs routinely identified in environmental extracts [90]. In this sense, since they also present adverse toxicological effects, they should be studied in more depth.

## 4. Conclusions

In conclusion, our results confirm the presence of PCBs in chinstrap penguins and krill from Deception Island. Specifically, the adult of *Pygoscelis antarticus* primarily accumulated PCBs in the liver and muscle, while the chicks exhibited the highest levels in their brains, and the PCB levels in the krill were lower than those found in krill from King George Island and the Ross Sea.

We also found high biomagnification factors for PCBs in our penguin samples, although the ∑PCBs levels in this study were lower than those found in seabirds from other locations in Antarctica and the Arctic. Finally, other-non-dioxin-like PCBs not covered by Directive 2013/39/EU and Commission Regulation (EU) No 277/2012 appeared predominantly in the analyzed penguins, indicating the need for further investigation in future research.

## Figures and Tables

**Figure 1 toxics-13-00430-f001:**
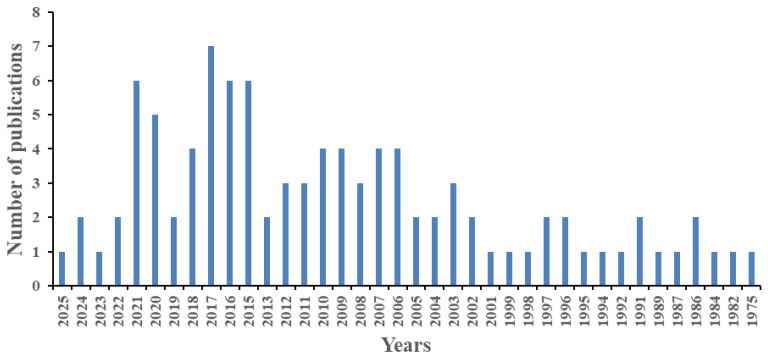
Publications on PCBs in penguins found in the WoS in May 2025.

**Figure 2 toxics-13-00430-f002:**
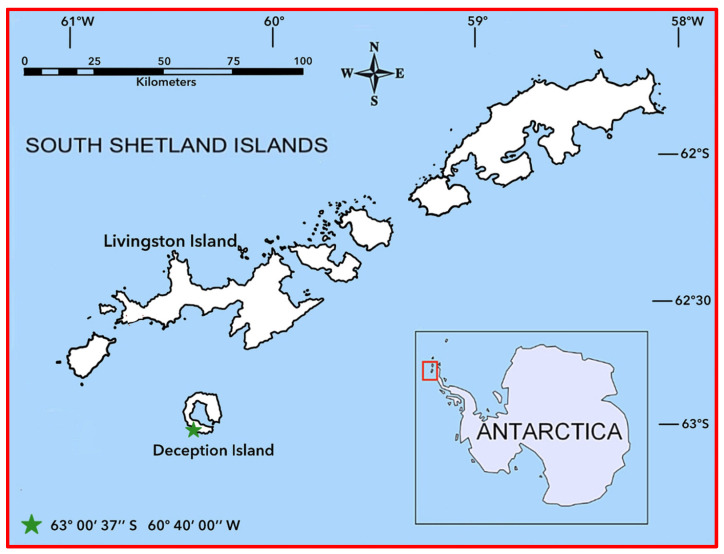
Map of the penguin breeding colony located at the South Shetland Islands (red square).

**Figure 3 toxics-13-00430-f003:**
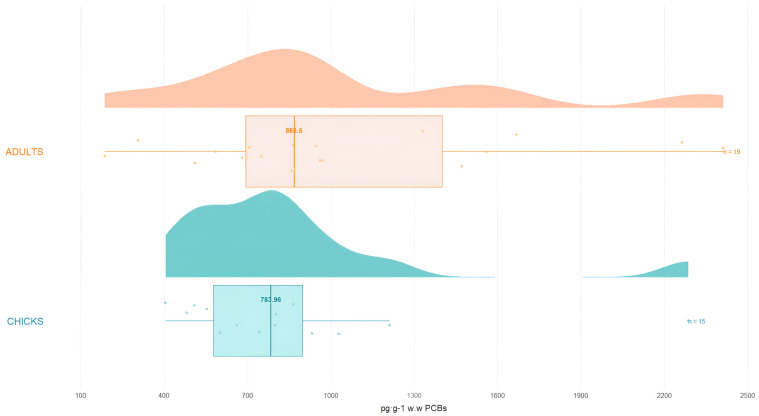
Total concentration (pg·g^−1^ w.w.) of PCBs in adults and chicks.

**Figure 4 toxics-13-00430-f004:**
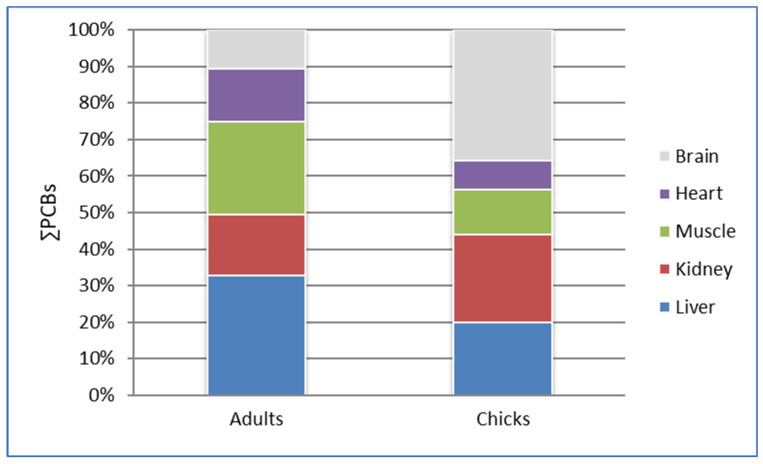
PCB percentage contribution in tissues of adult and chick chinstrap penguins (results on w.w. basis).

**Figure 5 toxics-13-00430-f005:**
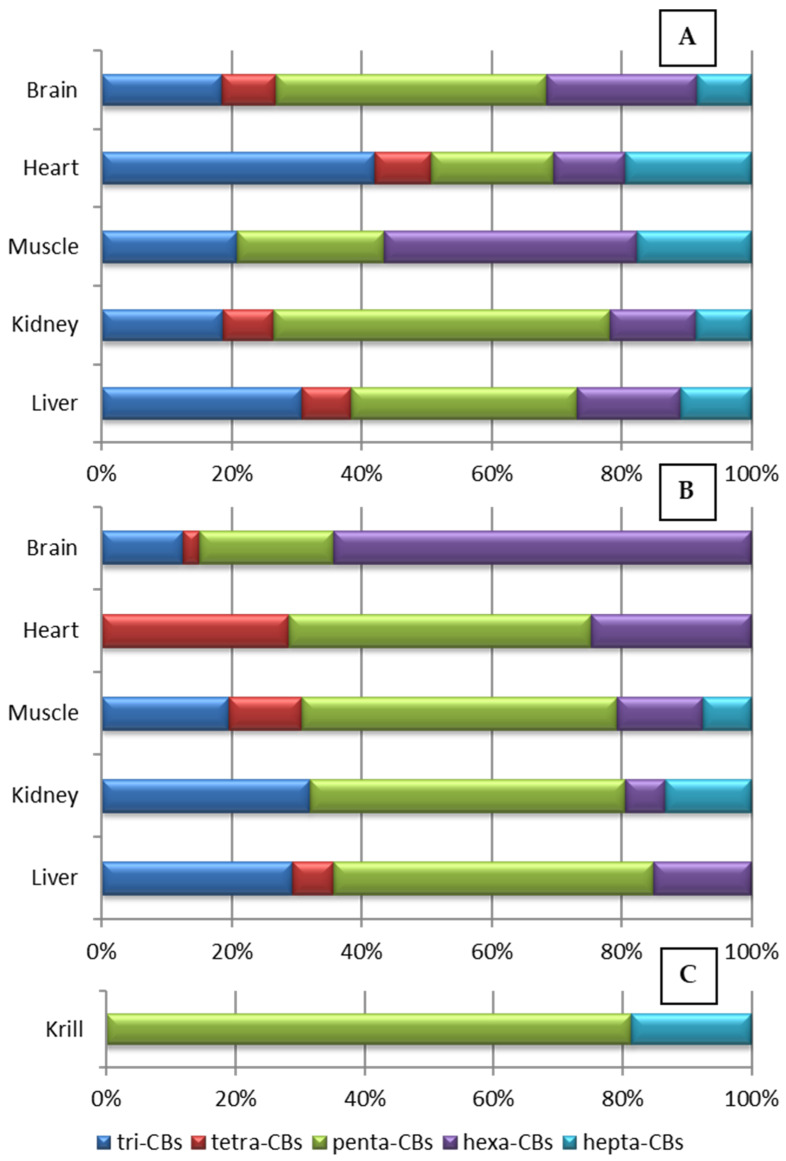
Classes of isomers in (**A**) adults, (**B**) chicks, and (**C**) krill (results on w.w. basis).

**Figure 6 toxics-13-00430-f006:**
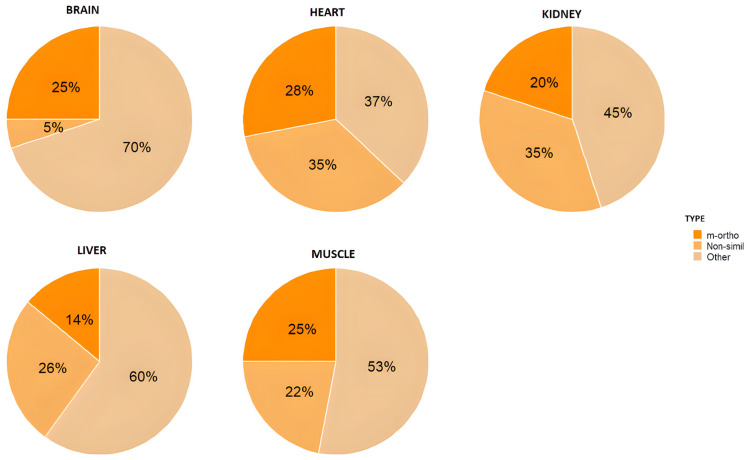
PCB distribution in tissues of adult chinstrap penguins.

**Figure 7 toxics-13-00430-f007:**
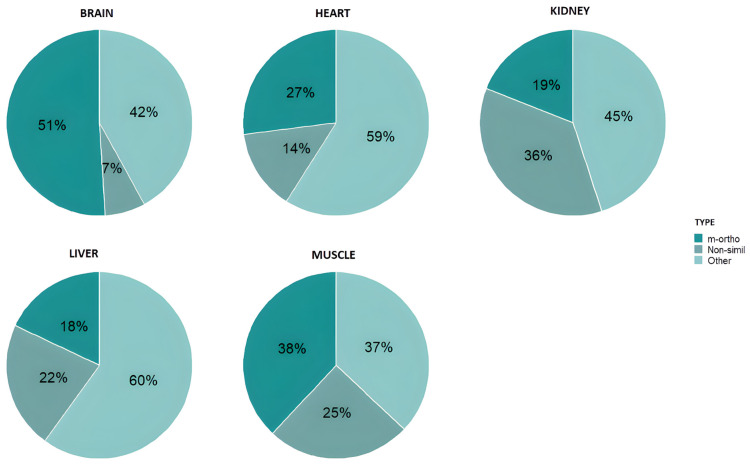
PCB distribution in tissues of chick chinstrap penguins.

**Table 1 toxics-13-00430-t001:** Studies on PCBs in different tissues of penguins in different years and locations.

Penguin Specie	Location	Sampled	Tissue	∑ PCBs	Reference
*Pygoscelis adeliae*	General Bernardo O′Higgins Chilean Military Base and Kopaitic Island (Antarctica)	2009	Feces	12,930 ± 2500 ^a^	[18]
*Pygoscelis antarctica*				4700 ± 1200 ^a^	
*Pygoscelis gentoo*				35,520 ± 38,450 ^a^	
*Pygoscelis antarctica*	King George Island (Antarctica)	2011	Feces	1450 ± 650 ^a^	[19]
*Pygoscelis adeliae*				1610 ± 470 ^a^	
*Pygoscelis papua*				2350 ± 760 ^a^	
*Spheniscus humboldti* (n = 29)	Punta San Juan (Perú)	2009	Blood	4590 ± 8240 ^a^	[20]
*Spheniscus humboldti* (n = 30)	Punta San Juan (Perú)	2011	Blood	1700–1750 ^b^	[21]
*Eudyptes chrysocome* (n = 17)	New Island (Falkland/Malvinas Islands)	2008/2009	Blood	550–1020 ^b^	[22]
			Eggs	25,800–27,800 ^c^	
*Pygoscelis adeliae* (n = 12)	Admiralty Bay, King George Island (Antarctica)	2004	Blood	9800 ± 3800 ^a^	[23]
*Pygoscelis antarctica* (n = 13)				4500 ± 2400 ^a^	
*Pygoscelis gentoo* (n = 16)				3400 ± 1600 ^a^	
*Eudyptes chrysocome* (n = 34)	New Island, (Falkland/Malvinas Islands)	2008	Eggs	27,550 ± 700 ^c^	[24]
*Pygoscelis adeliae* (n = 6)	Edmonson Point Rookery (Antarctica)	1995/1996	Eggs	24,900 ± 21,600 ^a^	[8]
*Euphausia superba* (krill)	Ross Sea (Antarctica)	2000 2001/2002	Wholebody	1670 ± 850 ^a^	
*Pygoscelis adeliae* (n = 5)	Ross Sea Terra Nova Bay (Antarctica)	1994/1995 1995/1996	Eggs	2800 ^a^	[25]
*Euphausia superba* (krill)			Wholebody	1900 ^a^	
*Pygoscelis adeliae* (n = 21)	Admiralty Bay, King George Island (Antarctica)	2010/2011 2011/2012	Eggs	57,300 ± 30,400 ^c^	[26]
*Pygoscelis papua* (n = 16)				44,310 ± 21,330 ^c^	
*Spheniscus demersus*	Robben Island (n = 10) (Africa)	2010/2011	Eggs	42,000 ± 12,000 ^a^	[27]
	Bird Island (n = 10) (Africa)			64,000 ± 16,000 ^a^	
*Pygoscelis adeliae* (n = 37)	Admiralty Bay, King George Island (Antarctica)	1995/1996 1998/1999 2000/2001 2001/2002 2004/2005	Eggs	4340 ± 2150 ^a^	[28]
*Pygoscelis emperor* (n = 6)				21,990 ± 25,580 ^a^	
*Pygoscelis adeliae* (n = 13)	Livingston Island, South Shetland, Antarctic Peninsula,	2004	Eggs	12,000 ± 4000 ^a^	[29]
*Pygoscelis papua* (n = 13)				5000 ± 3000 ^a^	
*Pygoscelis antarcticus* (n = 9)				6000 ± 4000 ^a^	
*Pygoscelis adeliae* (n = 3)	Admiralty Bay, King George Island, (Antarctica)	2004–2006	Eggs	32,500 ^a^	[30]
*Pygoscelis antarctica* (n = 26)				37,300 ^a^	
*Pygoscelis papua* (n = 9)				26,000 ^a^	
*Euphausia superba* (krill)				12,300 ^a^	
*Spheniscus magellanicus*	Região dos Lagos, Rio de Janeiro (Brazil)	2012	Muscle (n = 13)	<LOQ-1,500,000 ^d^	[31]
			Liver (n = 9)	<LOQ-1,163,000 ^d^	
*Spheniscus magellanicus* (n = 25)	Ubatuba, São Paulo (Brazil)	2008	Liver	18,900–775,800 ^e^	[32]
*Spheniscus magellanicus* (n = 116)	Six areas located in South America (Chile, Uruguay y Brasil)	2008–2012 except 2009	Liver	9900–818,000 ^a^ (2008) 203,000–835,000 ^a^ (2010) 13,300–456,000 ^a^ (2011) 500–492,000 ^a^ (2012)	[33]
*Pygoscelis adeliae* (n = 2) and *Pygoscelis papua* (n = 5) pooled together	Admiralty Bay, King George Island (Antarctica)	1997/1998	Fat	256,000 ± 125,000 ^c^	[34]
*Pygoscelis adeliae* (n = 4)	Admiralty Bay, King George Island (Antarctica)	2005/2006 2006/2007	Fat	114,000–325,000 ^a^	[35]
*Pygoscelis papua* (n = 2)				304,000–627,000 ^a^	
*Pygoscelis antarcticus* (n = 3)				221,000–550,000 ^a^	
*Pygoscelis papua* (n = 4)	Admiralty Bay, King George Island (Antarctica)	1991–1993	Brain	nd-7800 ^a^	[36]
			Liver	nd-1100 ^a^	
			Muscle	nd	
			Blood	2200–4800 ^a^	
			Bone	2100–16,500 ^a^	
			Uropygeal gland	48,200–1,047,300 ^a^	
			Fat	43,200–1,583,600 ^a^	
			Brain	4800 ^a^	
*Pygoscelis adeliae* (n = 1)			Liver	nd	
			Muscle	nd	
			Blood		
			Bone	32,100 ^a^	
			Uropygeal gland	77,300 ^a^	
			Fat	72,700 ^a^	
*Pygoscelis antarctica* (n = 15)	Cape Shirreff, King George Island (Antanrtica)	2012	Blood	80,024 ± 1240 ^a^	[37]
	Kopaitic Island, King George Island (Antanrtica)			7580 ± 900 ^a^	
	Narębski Poon, King George Island (Antanrtica)			7305 ± 1090 ^a^	
*Spheniscus demersus* (n = 21)	The Eastern Cape, South Africa (Africa)	1981/1983	Eggs	240 ^a^	[38]
*Pygoscelis adeliea* (n = 27)	Cape Bird, Ross Island (Antarctica)	1988/1989 1989/1990	Eggs	8800 ^a^	[39]
*Pygoscelis adeliae* (n = 5)	Terra Nova Bay (Antarctica)	1995/1996	Eggs	30,000 ^e^	[40]
*Euphausia superba* (krill)				900 ^e^	
*Pygoscelis papua* (n = 21)	King George Island (Antarctica)	2013/2014	Muscle	382–526,000 ^c^	[41]
*Pygoscelis antartica* (n = 8)				9050–124,000 ^c^	
*Pygoscelis papua* (n = 2)	Barton peninsula of King George Island (Antarctica)	2008/2009	Pectoralis	2506–5650 ^c^	[42]
*Pygoscelis adeliae* (n = 2)					
*Pygoscelis antárctica* (n = 15)	Cape Shirreff, Nar bski Point, and Kopaitic Island (Antarctica)	2013/2014	Blood	1200–2900 ^a^	[43]
*Pygoscelis antarticus* (n = 20)	Deception Island and Livingston Island (Antarctica)	2016–2017	Eggs	4710 ^c^	[16]
*Pygoscelis papua* (n = 10)				3200 ^c^	
*Pygosceli adeliae* (n = 13)	Admiralty Bay, King George Island (Antarctica)	2013–2014	Breast feathers	15,180 ± 9004 ^d^	[44]
*Pygoscelis antarcticus* (n = 14)				11,810 ± 4430 ^d^	
*Pygoscelis papua* (n = 14)				18,650 ± 5620 ^d^	
*Eudyptula minor* (n = 15)	Phillip Island, Victoria (Australia)	2018	Blood	12,900 ± 11,300 ^a^	[45]
*Pygoscelis adeliae* (n = 1)	Larsemann Hills, Prydz Bay (Antarctica)	2009	Composite	144,000 ^c^	[46]
*Aptenodytes forsteri* (n = 1)			Brisket	12,500 ^c^	
			Back leg fat	17,700 ^c^	
			Abdominal fat	17,400 ^c^	
			Breast fat	15,300 ^c^	
			Liver	6300 ^c^	
*Pygoscelis antarcticus* (n = 7)	South Shetland Islands (Antarctica)	2011/2012	Eggs	2110–5160 ^a^	[47]
*Pygoscelis papua* (n = 4)	Adeliae Island (Antarctica)	2001/2002	Eggs	500–800 ^e^	[48]
*Pygoscelis adeliae*	Adèlie Cove (Antarctica) (n = 8)	2018/2019 2021/2022	Eggs	20,900 ± 6640 ^a^	[49]
	Edmonson Point (Antarctica) (n = 5)			24,300 ± 6620 ^a^	
	Inexpressible Island (Antarctica) (n = 5)			22,600 ± 8870 ^a^	
*Pygoscelis adeliae* (n = 6)	Terra Nova Bay (Antarctica)			101,000 ^a^	[50]
*Pygoscelis adeliae* (n = 15)	Hop Island (Antarctica)	1993/1994	Blood	4660–5660 ^c^	[51]
			Uropygial oil		
*Pygoscelis adeliae* (n = 50)		1997/1998 2000/2001 2002/2003 2004/2005 2010/2011 2018/2019 2021/2022	Eggs		[52]
*Aptenodytes patagonicus* (n = 8)	Kamogawa city, Chiba Prefecture (Japan)	2020	Blood	19,100 ^b^	[53]
*Pygoscelis adeliae* (n = 24)	Hop Island (n = 8), Gardner Island (n = 8) and Rookery Lake (n = 8) (Antarctica)	2016/2017	Blood	61,100 ± 87,600 ^a^	[54]
*Pygoscelis adeliae* (n = 2)	Potter peninsula (Antarctica)	1993/1994	Eggs	200–300 ^a^	[55]
*Pygoscelis antarctica* (n = 3)				300–1500 ^a^	
*Pygoscelis papua* (n = 2)				100–300 ^a^	
*Aptenodytes forsteri*, *Pygoscelis adeliae*, *Eudyptes chrysocome*, *Eudyptes schlegeli*, and *Pygoscelis papua*	Davis and Casey stations and Macquarie Island (Antarctica)	1981–1983 1978–1983	Eggs	<100 ^a^	[56]

^a^: pg·g^−1^ wet weight; ^b^: pg·mL^−1^; ^c^: pg·g^−1^ lipid weight; ^d^: pg·g^−1^ dry weight; ^e^: pg·g^−1^; nd: none detected.

**Table 2 toxics-13-00430-t002:** Identification number of collected specimens (# sample), their estimated life stage (adult, and chick = pullus), weight (kg), collected tissues (L = liver, K = kidney, M = muscle, H = heart, and B = brain) (ND = not detected; and n.a. = not available), and, for chick samples, number of pools and pooled specimens. The weight (g) of pooled krill samples is also reported.

# Sample	Life Stage	Weight	Tissue
1A	adult	2.55	L, K, M, H
2A	adult	ND	L, K, M, H, B
3A	adult	3.50	L, K, M, H, B
4A	adult	ND	L, K, M, H, B
1C	pullus	2.65	L, K, M, H, B
2C	pullus	2.15	L, K, M, H, B
3C	pullus	ND	L, K, M, H, B
4C	pullus	1.85	L, K, M, H, B
5C	pullus	2.00	L, K, M, H, B
6C	pullus	ND	L, K, M, H, B
Chicks			
tissue	no. of pool	no. of specimen	
liver	3	1C + 6C, 3C + 4C, 2C + 5C	
kidney	3	1C + 2C, 3C + 4C, 5C + 6C	
muscle	5 single sample	(sample no. 4C: n.a.)	
heart	2	1C + 2C + 3C, 4C + 5C + 6C	
brain	2	1C + 2C + 3C, 4C + 5C + 6C	
Krill	5.1127 g		

**Table 3 toxics-13-00430-t003:** Concentrations of PCBs (average ± SD and min–max in pg·g^−1^ w.w. and lipid basis * and percentage of detectable levels; n = 34) in tissues of chinstrap penguins and krill from Deception Island.

**Samples**	**28**	**(%)**	**52**	**(%)**	**37**	**(%)**	**95**	**(%)**	**101**	**(%)**
Liver (A)	87.174 ± 172.442 <1.907–345.837	(25)	100.788 ± 58.821 31.620–153.568	(100)	322.266 ± 403.934 <1.842–839.107	(50)	112.249 ± 49.122 70.281–183.022	(100)	73.673 ± 87.445 <1.426–175.328	(50)
Liver (C)	71.101 ± 121.499 <1.907–211.395	(33)	41.617 ± 14.519 27.044–56.081	(100)	126.986 ± 218.352 <1.842–379.118	(33)	180.245 ± 96.753 111.150–290.822	(100)	39.956 ± 67.972 <1.426–118.444	(33)
Kidney (A)	126.191 ± 250.475 <1.907–501.903	(25)	52.545 ± 54.678 <0.199–119.486	(75)	<1.842	(0)	139.039 ± 51.894 85.189–209.574	(100)	32.616 ± 63.806 <1.426–128.325	(25)
Kidney (C)	125.103 ± 107.873 <1.907–195.941	(66)	<0.199	(0)	135.592 ± 233.257 <1.842–404.934	(33)	118.081 ± 204.511 <0.012–354.230	(33)	59.088 ± 101.109 <1.426–175.839	(33)
Muscle (A)	64.199 ± 126.490 <1.907–253.934	(25)	<0.199	(0)	151.035 ± 300.228 <1.842–601.376	(25)	74.574 ± 55.421 <0.012–126.569	(75)	45.247 ± 89.068 <1.426–178.849	(25)
Muscle (C)	80.776 ± 109.644 <1.907–212.753	(40)	46.299 ± 103.306 <0.199–231.100	(20)	<1.842	(0)	92.548 ± 131.903 <0.012–319.631	(60)	<1.426	(0)
Heart (A)	245.137 ± 217.542 <1.907–530.572	(75)	51.394 ± 44.296 <0.199–107.512	(75)	<1.842	(0)	28.185 ± 56.358 <0.012–112.722	(25)	<1.426	(0)
Heart (C)	<1.907	(0)	74.645 ± 54.542 36.078–113.212	(100)	<1.842	(0)	<0.012	(0)	<1.426	(0)
Brain (A)	<1.907	(0)	35.404 ± 61.149 <0.199–106.012	(33)	80.086 ± 137.119 <1.842–238.417	(33)	<0.012	(0)	<1.426	(0)
Brain (C)	<1.907	(0)	31.903 ± 44.977 <0.199–63.706	(50)	151.702 ± 213.237 <1.842–302.483	(50)	112.971 ± 66.868 65.689–160.254	(100)	52.025 ± 72.566 <1.426–103.337	(50)
Krill	<1.907	(0)	<0.199	(0)	<1.842	(0)	144.233	(100)	<1.426	(0)
**Samples**	**99**	**(%)**	**110**	**(%)**	**151**	**(%)**	**149**	**(%)**	**123**	**(%)**
Liver (A)	110.939 ± 101.227 <0.002–219.834	(75)	131.032 ± 41.309 89.957–167.070	(100)	22.350 ± 31.892 <0.520–67.885	(50)	14.481 ± 21.750 <0.011–46.002	(50)	<1.066	(0)
Liver (C)	<0.002	(0)	70.280 ± 67.952 <0.012–135.644	(66)	<0.520	(0)	40.684 ± 49.534 <0.011–95.846	(66)	<1.066	(0)
Kidney (A)	27.264 ± 35.091 <0.002–73.516	(50)	59.897 ± 61.867 <0.012–146.243	(75)	11.936 ± 23.352 <0.520–46.963	(25)	5.994 ± 11.977 <0.011–23.959	(25)	<1.066	(0)
Kidney (C)	26.720 ± 46.278 <0.002–80.158	(33)	29.988 ± 40.883 <0.012–76.558	(66)	12.628 ± 12.960 <0.520–26.109	(66)	6.142 ± 10.628 <0.011–18.414	(33)	<1.066	(0)
Muscle (A)	43.910 ± 50.718 <0.002–89.369	(50)	5.700 ± 11.389 <0.012–22.783	(25)	33.475 ± 47.326 <0.520–100.647	(50)	29.331 ± 46.861 <0.011–99.239	(75)	<1.066	(0)
Muscle (C)	28.604 ± 39.747 <0.002–95.664	(60)	<0.012	(0)	<0.520	(0)	<0.011	(0)	<1.066	(0)
Heart (A)	42.504 ± 85.006 <0.002–170.013	(25)	<0.012	(0)	<0.520	(0)	6.336 ± 12.660 <0.011–25.325	(25)	<1.066	(0)
Heart (C)	30.040 ± 42.481 <0.002–60.079	(50)	43.637 ± 61.703 <0.012–87.268	(50)	<0.520	(0)	<0.011	(0)	<1.066	(0)
Brain (A)	35.045 ± 34.218 <0.002–68.373	(66)	78.745 ± 136.379 <0.012–236.222	(33)	<0.520	(0)	21.785 ± 37.723 <0.011–65.343	(33)	<1.066	(0)
Brain (C)	<0.002	(0)	<0.012	(0)	37.468 ± 52.621 <0.520–74.677	(50)	<0.011	(0)	<1.066	(0)
Krill	35.551	(0)	<0.012	(0)	<0.520	(0)	<0.011	(0)	<1.066	(0)
**Samples**	**118**	**(%)**	**114**	**(%)**	**146**	**(%)**	**153**	**(%)**	**105**	**(%)**
Liver (A)	<0.427	(0)	<0.003	(0)	<0.011	(0)	91.611 ± 69.333 41.784–193.596	(100)	34.656 ± 30.576 <0.462–61.992	(75)
Liver (C)	33.953 ± 29.246 <0.427–52.067	(66)	<0.003	(0)	22.491 ± 38.945 <0.011–67.461	(33)	37,787 ± 33,551 <2.541–67.251	(75)	6.870 ± 11.500 <0.462–20.149	(33)
Kidney (A)	91.532 ± 39.741 50.726–146.019	(100)	<0.003	(0)	9.152 ± 18.292 <0.011–36.590	(25)	<2.541	(0)	<0.462	(0)
Kidney (C)	56.284 ± 52.180 <0.427–103.418	(66)	88.570 ± 153.405 <0.003–265.706	(33)	<0.011	(0)	16.739 ± 26.792 <2.541–47.676	(33)	16.005 ± 27.322 <0.462–47.555	(33)
Muscle (A)	<0.427	(0)	46.914 ± 93.825 <0.003–187.652	(25)	20.994 ± 26.923 <0.011–56.344	(50)	197.599 ± 392.657 <2.541–786.585	(25)	14.972 ± 29.483 <0.462–59.197	(25)
Muscle (C)	62.375 ± 116.234 <0.427–267.556	(40)	16.715 ± 37.373 <0.003–83.569	(20)	<0.011	(0)	41.148 ± 43.891 <2.541–100.448	(60)	<0.462	(0)
Heart (A)	9.168 ± 17.908 <0.427–36.030	(25)	29.691 ± 43.883 <0.003–92.931	(50)	<0.011	(0)	14.235 ± 25.928 <2.541–53.127	(25)	<0.462	(0)
Heart (C)	<0.427	(0)	47.115 ± 66.629 <0.003–94.230	(50)	<0.011	(0)	64.045 ± 18.763 50.778–77.312	(100)	<0.462	(0)
Brain (A)	<0.427	(0)	65.531 ± 57.007 <0.003–103.699	(66)	<0.011	(0)	<2.541	(0)	<0.462	(0)
Brain (C)	7.027 ± 9.636 <0.427–13.841	(0)	51.881 ± 73.369 <0.003–103.761	(50)	<0.011	(0)	320.642 ± 451.660 <2.541–640.014	(50)	25.028 ± 35.068 <0.462–49.825	(50)
Krill	<0.427	(0)	<0.003	(0)	<0.011	(0)	<2.541	(0)	<0.462	(0)
**Samples**	**138**	**(%)**	**187**	**(%)**	**183**	**(%)**	**128**	**(%)**	**167**	**(%)**
Liver (A)	79.363 ± 86.872 <0.010–194.561	(75)	<0.430	(0)	<0.003	(0)	2.563 ± 4.032 <1.093–8.610	(25)	<0.010	(0)
Liver (C)	<0.010	(0)	<0.430	(0)	<0.003	(0)	<1.093	(0)	<0.010	(0)
Kidney (A)	62.475 ± 77.945 <0.010–161.112	(50)	21.867 ± 33.130 <0.430–70.142	(50)	<0.003	(0)	<1.093	(0)	<0.010	(0)
Kidney (C)	13.123 ± 22.722 <0.010–39.360	(33)	<0.430	(0)	<0.003	(0)	<1.093	(0)	<0.010	(0)
Muscle (A)	118.561 ± 150.871 <0.010–333.966	(75)	<0.430	(0)	<0.003	(0)	<1.093	(0)	<0.010	(0)
Muscle (C)	12.591 ± 28.144 <0.010–62.937	(20)	11.839 ± 25.991 <0.430–58.332	(20)	<0.003	(0)	<1.093	(0)	<0.010	(0)
Heart (A)	<0.010	(0)	<0.430	(0)	<0.003	(0)	<1.093	(0)	<0.010	(0)
Heart (C)	<0.010	(0)	<0.430	(0)	<0.003	(0)	<1.093	(0)	<0.010	(0)
Brain (A)	<0.010	(0)	<0.430	(0)	<0.003	(0)	28.442 ± 48.316 <1.093–84.233	(33)	50.116 ± 86.795 <0.010–150.339	(33)
Brain (C)	<0.010	(0)	<0.430	(0)	<0.003	(0)	<1.093	(0)	150.220 ± 212.436 <0.010–300.435	(50)
Krill	<0.010	(0)	40.891	(0)	<0.003	(0)	<1.093	(0)	<0.010	(0)
**Samples**	**177**	**(%)**	**156**	**(%)**	**157**	**(%)**	**180**	**(%)**	**170**	**(%)**
Liver (A)	<0.003	(0)	<1.845	(0)	<0.010	(0)	<0.003	(0)	145.761 ± 291.519 <0.003–583.039	(25)
Liver (C)	<0.003	(0)	<1.845	(0)	<0.010	(0)	<0.003	(0)	<0.003	(0)
Kidney (A)	<0.003	(0)	<1.845	(0)	<0.010	(0)	<0.003	(0)	35.611 ± 71.219 <0.003–142.440	(25)
Kidney (C)	<0.003	(0)	<1.845	(0)	<0.010	(0)	108.207 ± 127.177 <0.003–248.291	(66)	<0.003	(0)
Muscle (A)	180.663 ± 361.323 <0.003–722.648	(25)	<1.845	(0)	<0.010	(0)	<0.003	(0)	<0.003	(0)
Muscle (C)	<0.003	(0)	<1.845	(0)	<0.010	(0)	19.159 ± 42.836 <0.003–95.787	(20)	<0.003	(0)
Heart (A)	52.975 ± 105.948 <0.003–211.897	(25)	43.606 ± 85.368 <1.845–171.658	(25)	<0.010	(0)	<0.003	(0)	60.454 ± 120.905 <0.003–241.812	(25)
Heart (C)	<0.003	(0)	<1.845	(0)	<0.010	(0)	<0.003	(0)	<0.003	(0)
Brain (A)	<0.003	(0)	<1.845	(0)	<0.010	(0)	<0.003	(0)	35.756 ± 61.929 <0.003–107.265	(33)
Brain (C)	<0.003	(0)	<1.845	(0)	271.753 ± 384.310 <0.010–543.501	(50)	<0.003	(0)	<0.003	(0)
Krill	<0.003	(0)	<1.845	(0)	<0.010	(0)	<0.003	(0)	<0.003	(0)
**Samples**	**189**		**(%)**	**209**		**(%)**	**ΣPCBs**		**ΣPCBs ***	
Liver (A)	<0.007		(0)	<0.007		(0)	1330.819 ± 733.689 730.216–2252.976		60,292.998 ± 33,239.914 33,082.569–102,071.490	
Liver (C)	<0.007		(0)	<0.007		(0)	674.478 ± 177.533 476.717–820.113		21,008.794 ± 5529.838 14,848.889–25,545.066	
Kidney (A)	<0.007		(0)	<0.007		(0)	680.565 ± 191.311 491.279–914.887		21,741.224 ± 6111.592 15,694.323–29,226.838	
Kidney (C)	<0.007		(0)	<0.007		(0)	814.613 ± 383.012 421.575–1186.743		19,584.246 ± 9208.055 10,135.154–28,530.685	
Muscle (A)	<0.007		(0)	<0.007		(0)	1029.727 ± 823.394 285.248–2150.527		74,343.653 ± 59,446.939 20,594.175–15,5262.543	
Muscle (C)	<0.007		(0)	<0.007		(0)	416.220 ± 252.869 144.322–762.238		22,630.902 ± 13,749.108 7847.141–41,444.749	
Heart (A)	<0.007		(0)	<0.007		(0)	587.141 ± 407.879 140.375–1056.493		40,068.400 ± 27,834.982 9579.644–72,098.498	
Heart (C)	<0.007		(0)	<0.007		(0)	265.036 ± 121.631 179.030–351.042		10,575.446 ± 4853.311 7143.641–14,007.251	
Brain (A)	<0.007		(0)	<0.007		(0)	436.254 ± 234.244 290.773–706.470		10,255.191 ± 5506.464 6835.313–16,607.262	
Brain (C)	<0.007		(0)	<0.007		(0)	1215.829 ± 955.189 540.408–1891.249		58,279.209 ± 45,785.764 25,903.767–90,654.603	
Krill	<0.007		(0)	<0.007		(0)	227.384		5535.934	

A: adult; C: chick; non-detectable levels were shown as <DL (detection limit).

**Table 4 toxics-13-00430-t004:** Concentrations of mono-*ortho* PCBs (average ± SD, min-max in pg·g^−1^ w.w.) in chinstrap penguins from Deception Island and the corresponding TEQ concentrations.

Mono-*ortho* PCBs	Concentrations	TEQ Concentrations
PCB 105	Liver adults 34.656 ± 30.576, <0.462–61.992 Brain chicks 25.028 ± 35.068, <0.462–49.825	0.00003
PCB 114	Brain chicks 51.881 ± 73.369, <0.003–103.761	0.00003
PCB 118	Brain chicks 7.027 ± 9.636, <0.427–13.841	0.00003
PCB 157	Brain chicks 271.753 ± 384.310, <0.010–543.501	0.00003
PCB 167	Brain chicks 150.220 ± 212.436, <0.010–300.435	0.00003

## Data Availability

The data presented in this study are available upon request from the corresponding author. The data are not publicly available due to privacy restrictions.

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
