# Peer review of "PCBs in Chinstrap Penguins from Deception Island (South Shetland Islands, Antarctica)"

_toxics, 2025, doi:10.3390/toxics13060430_

Round 1
Reviewer 1 Report
Comments and Suggestions for Authors
This study measured the concentrations of PCBs in penguin tissues and krill from Deception Island, and assessed the toxic potential of PCBs using TEQ values. It provides valuable data for the study of PCBs in Antarctica, confirms the presence of PCBs in chinstrap penguins and krill in the Deception Island area, and reveals the distribution patterns of PCBs in penguin tissues. The findings are meaningful and have publication value. However, the following issues need to be addressed before publication:
- The Introduction mentions a literature search and presents a publication volume chart covering the period from 1975 to 2024. However, I believe this statistic has limited significance. The authors may need to focus more on content that is closely related to the theme of the paper, such as historical changes in PCB concentrations in penguin tissues reported in the literature. This would be more meaningful and relevant to the study.
- In line 134, the article states that the samples used for analysis were collected between 2007 and 2010. The time gap between sample collection and analysis appears to be quite long. Please clarify whether this delay could affect the accuracy of the measurements.
- In lines 209–210, “such as” is incorrectly written as “such us” and should be corrected. Please check the manuscript for other spelling errors and make the necessary corrections.
- The current Results section contains substantial content that aligns more with a discussion. Therefore, it is recommended that the section title be revised to “Results and Discussion” to more accurately reflect its content. Additionally, there is an error in the section numbering of the Conclusion—it should be labeled as Section 4 rather than Section 5.
- Figures 6 and 7 are not clear, and the text within the figures is too small. Please improve the image quality and enlarge the text to ensure that readers can easily extract the relevant information from the figures.
- Section 3.2 in the Results appears to be relatively brief and reads more like a summary of data from other studies rather than an in-depth analysis. It is recommended that the authors expand the discussion in this section to provide more insightful interpretation and context.
Reviewer 2 Report
Comments and Suggestions for Authors
General Comments:
The manuscript addresses an important and interesting topic concerning PCB contamination in Antarctic wildlife, particularly in chinstrap penguins and their prey, krill. However, the statistical analysis and data presentation require significant improvement before the manuscript can be considered for publication. The overall quality of the statistical evaluation, figures, and tables needs major revision to meet high scientific standards. Below, I outline specific recommendations for improvement.
Major Points:
-
Statistical Analysis:
-
The manuscript employs parametric tests (e.g., ANOVA, t-test) without verifying whether the assumptions underlying these tests were met.
-
Before applying parametric tests, the authors must:
-
Test for normality (e.g., Shapiro-Wilk test).
-
Test for homogeneity of variances (e.g., Levene’s test).
-
-
Only if the data meet the assumptions (normality and equal variances) can parametric tests be used.
-
If the assumptions are violated, non-parametric alternatives (e.g., Kruskal-Wallis test, Mann-Whitney U test) must be applied.
-
Without verifying these assumptions, the reliability of the statistical conclusions is seriously questionable.
-
-
Table 2 Presentation:
-
Table 2 is extremely dense, complicated, and overwhelming.
-
It is very difficult for readers to maintain focus or interpret the results.
-
I strongly recommend:
-
Reformatting Table 2: Organize it into clearer sections or split it into multiple, simpler tables.
-
Using graphical presentations, such as bar charts or box plots, would be more effective in visualizing differences between tissues and between adults and chicks.
-
-
Furthermore, no statistical comparisons are shown in the table — this is a major flaw. Appropriate statistical indicators (e.g., superscript letters for group comparisons, p-values) must be added.
-
-
Figures 3–5:
-
Figures 3–5 lack statistical comparisons.
-
Statistical differences (e.g., significance letters, p-values, error bars) should be shown clearly in the graphs.
-
-
Figures 6–7:
-
Figures 6 and 7 are of poor visual quality. They are difficult to read and interpret
-
-
Recommended Literature:
To improve their statistical approach, interpretation, and presentation of results, I recommend that the authors review and cite the following papers:
-
https://www.nature.com/articles/s41598-024-61986-4
-
https://www.sciencedirect.com/science/article/pii/S1532045622001818
These papers provide excellent examples of appropriate statistical testing, clear graphical representation of biological data, and meaningful biological interpretation of differences.
-
Minor Points:
-
Typographical errors (minor typos were noticed; recommend thorough proofreading).
-
Pay attention to consistent use of units (e.g., pg·g⁻¹ w.w.) throughout the text.
Round 2
Reviewer 2 Report
Comments and Suggestions for Authors
The authors have thoroughly addressed all of the reviewer’s comments and suggestions, and they have revised the manuscript accordingly. I find the revisions to be appropriate and satisfactory. Therefore, I consider the manuscript suitable for publication in Toxics.